# Quasi-Stable, Non-Magnetic, Toroidal Fluid Droplets in a Ferrofluid with Annular Magnetic Field

**Alastair Radcliffe**

Department Maths, Warwick University, Coventry CV4 8UW, UK; alastair.radcliffe@gmail.com

**Abstract:** A relatively stable, non-magnetic, torus-shaped fluid droplet within a linearly magnetizable surrounding ferrofluid medium, and subject to the annular magnetic field induced by an electric current in a wire passing perpendicularly through its centre, has been found through the use of coupled finite element/boundary element computer simulations.

**Keywords:** ferrofluids; toroidal droplets; annular magnetic fields; finite/boundary elements



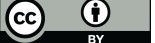

## 1. Introduction

A ferrofluid may be thought of as a "liquid magnet", with the addition of a surfactant to a simple Newtonian carrier fluid allowing the stable colloidal suspension of magnetic or magnetizable particles that make this fluid sensitive to the presence of a magnetic field [1].

External magnetic fields may then be used to manipulate and control the surfaces and interfaces of these liquids in a non-invasive manner, giving rise to a host of different engineering and industrial applications.

If these surfaces or interfaces are those of single, isolated droplets, perhaps sitting on a substrate surface in air or contained within another immiscible fluid, then such magnetic fields may be used to control both the motion and shape of the whole droplets themselves.

Such drops may then be used to transport various chemical species dissolved, or "encapsulated", within them to different locations in a "lab-on-a-chip" for analysis, say, or act as "microreactors" when their contents are actuated or agitated by some external magnetic stimuli; see [2] and the references therein.

While the majority of research into ferrofluids and droplets has involved the droplet being composed of the ferrofluid, for example in numerous extension experiments of the type seen in [3], the inverse situation has more rarely been studied with, for example, the deformation of aqueous droplets in an oil-based ferrofluid [4], or of oil drops in an aqueous ferrofluid [5], and the splitting and demulsification of micro-droplets within a hydrophobic ferrofluid continuous phase [6].

The use of toroidal, or "ring" shaped, annular droplets in ferrofluidics is also fairly limited, but their dynamics under a magnetic field has been considered in a Hele–Shaw cell [7] (the fluid region between two parallel glass plates separated by a narrow gap), in single droplet soft robotics [8], and for manipulating isolated water droplets on a substrate [9]—albeit always with the annular droplet being that composed of the ferrofluid, and thus giving, perhaps, some novelty to the present work where it is the surrounding medium that is a ferrofluid.

## 2. Methods

### 2.1. Numerical Implementation

Consider a Newtonian, viscous, non-magnetic fluid droplet, occupying an "interior" region $\Omega$, with a density, $\rho$, and dynamic viscosity, $\nu$—excuse the non-standard notation, but $\mu$ will be needed elsewhere—so much greater (for convenience) than that of its linearly magnetic ferrofluid surrounding "exterior" medium, occupying $\Omega^\infty$, and with which it shares a boundary $\Gamma$, that the latter's density and viscosity may be neglected.

Given that quasi-stable equilibrium geometries will be of primary interest, using *non*-negligible surrounding ("matrix") fluid parameters would only effect the kinematics of actually reaching any stable state, and not the state itself. Thus the extra calculations needed for simulating the ambient fluid motions in the (unbounded) exterior would be superfluous—a similar argument could be made against the interior fluid too, but simulating the fluid parameters here does at least allow good control of the progression of the calculations to the steady state.

With a surface energy/tension of $\gamma$, between the interior and exterior fluids, the fully coupled governing equations, derived from [3] and all collected together for ease of reference in the next sub-section, will start with those of incompressible Navier-Stokes (5a) and (5b), for the interior fluid velocities, **u**, and pressure, $p$.

The divergence of the magnetic stress tensor, $\nabla \cdot \boldsymbol{\sigma}_m$, now present only in the exterior rather than the interior [10], then gives rise to the extra magnetic forcing—after application of the divergence theorem to its integral—to be added to the usual local mean curvature, $\kappa$, based surface tension forcing (5h).

Now, the magnetic field, **H**, and inductive flux, **B**, in an electrically non-conducting medium are, of course, governed by Maxwell's equations. However, these equations are not here discretised directly, but their representations on the "magnetic vector potential", or "MVP", are.

Before introducing the two MVP's used, let the toroidal droplet lye in a Cartesian $x - y$ plane, with the current carrying wire coincident with the $z$ axis through its centre, and $(\cdot, \cdot, \cdot)$ indicate a simple three-dimensional Cartesian vector triple; it is also useful to adopt the terminology of wave scattering problems for describing the various magnetic fields present.

Thus a 'total' (superscript 't') magnetic field is the summation of an 'incident' (subscript '0') field hitting the target torus droplet and the 'scattered' (superscript 's') field purely arising due to the droplet's presence—if, later on, no clarifying super-/sub-script is present, then a total value is to be assumed.

Because of this simple geometrical setup, the magnetic fields can then be approximately described everywhere by the single, scalar, 'z'-component of either a scattered or a total magnetic vector potential $\mathbf{A}^{s|t}$.

The single, scalar MVP component for the *total* magnetic field inside the *non*-magnetisable drop, is given by $\phi$, while that for only the *scattered* field in the linearly magnetisable outside medium is $\psi$, such that $\mathbf{A}^t = (0, 0, \phi)$ inside, $\mathbf{A}^s = (0, 0, \psi)$ outside and, by definition of the MVP, $\nabla \times \mathbf{A}^{s|t} = \mathbf{H}^{s|t}$ everywhere.

Of course, this is a notable simplification of the "true" MVP for the three-dimensional problem considered here; however, given the relatively flat nature of the toroidal geometries to be studied, the central premise of the current work is that informative results may still be obtained when any $x$ or $y$ components to the MVP are neglected.

This all allows respective expressions for the total magnetic fields inside, $\mathbf{H}^t = (\phi_y, -\phi_x, 0)$ and outside, $\mathbf{H}^t = \mathbf{H}_0 + (\psi_y, -\psi_x, 0)$, where $\mathbf{H}_0$ is the imposed incident magnetic field and a subscript $x$, $y$ or $z$ ONLY ON $\phi$ or $\psi$ indicates a derivative of $\phi$ or $\psi$ in that Cartesian direction.

The imposed magnetic field of interest here is that which decreases with radial, $r$, distance from an infinitely long straight wire, of negligible thickness, directed along the Cartesian $z$-axis, centred at the origin, and carrying an electric current $I_0$

$$\mathbf{H}_0 = \frac{I_0}{2\pi} \left( \frac{y}{r^2}, \frac{-x}{r^2}, 0 \right) = \frac{I_0}{2\pi} \left( \frac{y}{x^2 + y^2}, \frac{-x}{x^2 + y^2}, 0 \right) \tag{1}$$

which may itself be formed from the curl of an MVP with a single scalar $z$-component, $h_0$, given by

$$h_0 = \frac{I_0}{4\pi} \log\left( x^2 + y^2 \right) \tag{2}$$

such that $\mathbf{H}_0 = \nabla \times (0, 0, h_0)$.

Now using the Coulomb gauge condition $\nabla \cdot \mathbf{A} \equiv 0$ together with the vector identity

$$\nabla \times \nabla \times \mathbf{A} = \nabla(\nabla \cdot \mathbf{A}) - \Delta\mathbf{A} \tag{3}$$

and letting $\mu_0$, and $\mu$, denote the magnetic permeability of the drop and that of its surrounding magnetizable medium respectively—so $\mathbf{B} = \mu_0\,\mathbf{H}$ inside the drop, and $\mathbf{B} = \mu\,\mathbf{H}$, outside—the Maxwell equations

$$\nabla \cdot \mathbf{B} = 0 \quad and \quad \nabla \times \mathbf{H} = \mathbf{0} \tag{4}$$

reduce to just the Laplace equation for the single MVP scalar components, both inside (5c), and outside (5d), where the latter has been cast into its boundary integral form [11] involving the use of the appropriate 3-D Greens function $G = 1/|\mathbf{r} - \mathbf{r}'|$ between spatial positions $\mathbf{r}$ and $\mathbf{r}'$.

Now, because we are assuming a linearly magnetisable surrounding matrix fluid, its magnetisation, $\mathbf{M}$, will be colinear with the applied external magnetic field, with a magnitude ratio given by the magnetic susceptibility, $\chi = (\mu/\mu_0 - 1)$, such that $\mathbf{M} = \chi\mathbf{H}$ and $\mu = \mu_0(\chi + 1)$.

So while the magnetic flux inside is just $\mathbf{B} = \mu_0\mathbf{H} = \mu_0(\phi_y, -\phi_x, 0)$, outside it is then the sum of this magnetisation and the incident field already there

$$\mathbf{B} = \mu_0(\mathbf{H} + \mathbf{M}) = \mu_0(\mathbf{H} + \chi\mathbf{H}) = \mu_0(1 + \chi)\mathbf{H} = \mu_0(1 + \chi)\big[\mathbf{H}_0 + (\psi_y, -\psi_x, 0)\big]$$

Across the surface, $\Gamma = \partial\Omega$, of the droplet, with normal $\mathbf{n} = (n_x, n_y, n_z)$, continuity of the normal magnetic flux, $\mathbf{B} \cdot \mathbf{n}$, and of the tangential magnetic field, $\mathbf{H} \times \mathbf{n}$, demand *jumps* in the MVPs (5e) and their normal derivatives (5f) due to the incident field.

Note that inside $\mathbf{H} \times \mathbf{n} = (\phi_y, -\phi_x, 0) \times (n_x, n_y, n_z) = (-\phi_x n_z, -\phi_y n_z, \phi_x n_x + \phi_y n_y)$ $= (\cdot, \cdot, \nabla\phi \cdot \mathbf{n}) = (\cdot, \cdot, \phi_n)$, and similarly for $\psi$ outside, when considering the gauge condition $\nabla \cdot \mathbf{A} \equiv 0$ attempted, such that $\phi_z \approx \psi_z \approx 0$.

For the exterior region, $\Omega^\infty$, of unlimited extent, an integral representation (the "boundary element method") of the MVP is used [11], requiring a solution only along the fluid droplet/medium interface $\Gamma$ to support values of the MVP either throughout $\Omega^\infty$ or else just on the interface $\Gamma$ itself; hence the integral form of the Laplace equation in (5d) involving either $4\pi$ or $2\pi$ as coefficients respectively. Such integral forms naturally support solutions that decay towards infinity (5g), leaving just the imposed magnetic field there.

### 2.2. Governing Equations

The complete set of equations then looks like:

$$\rho\frac{\partial\mathbf{u}}{\partial t} + \rho(\mathbf{u} \cdot \nabla)\mathbf{u} = \qquad\qquad -\nabla p - \nu\Delta\mathbf{u} \qquad\qquad \mathbf{u}, p \in \Omega \tag{5a}$$

$$\nabla \cdot \mathbf{u} = \qquad\qquad 0 \qquad\qquad \mathbf{u} \in \Omega \tag{5b}$$

$$\Delta\phi = \qquad\qquad 0 \qquad\qquad \phi \in \Omega \tag{5c}$$

$$(4\pi \mid 2\pi)\,\psi = \qquad \oint_\Gamma G\frac{\partial\psi}{\partial n}\,ds - \oint_\Gamma \psi\frac{\partial G}{\partial n}\,ds \qquad \psi \in \Omega^\infty \mid \psi \in \Gamma \tag{5d}$$

$$\phi - (1+\chi)\,\psi = \qquad\qquad (1+\chi)\,h_0 \qquad\qquad \phi, \psi \in \Gamma \tag{5e}$$

$$\frac{\partial\phi}{\partial n} - \frac{\partial\psi}{\partial n} = \qquad\qquad \frac{\partial h_0}{\partial n} \qquad\qquad \phi, \psi \in \Gamma \tag{5f}$$

$$\lim_{|r|\to\infty}\psi = \qquad\qquad 0 \qquad\qquad \psi \in \Omega^\infty \tag{5g}$$

$$\frac{\partial\mathbf{u}}{\partial n} = \qquad\qquad 2\gamma\kappa\,\mathbf{n} + \boldsymbol{\o}_m \cdot \mathbf{n} \qquad\qquad on\ \Gamma \tag{5h}$$

where $\nu$ denotes the dynamic viscosity, and $\boldsymbol{\sigma}_m$ the magnetic stress tensor for an incompressible, isothermal, linearly magnetizable medium, which, if the magnetic field **H** has scalar magnitude $H$, and $I$ represents the identity matrix, is given by [3]

$$\boldsymbol{\sigma}_m = -\frac{\mu}{2}H^2 I + \mu\,\mathbf{H}\,\mathbf{H}^T \tag{6}$$

Note that in [10] the *gradient* of a scalar potential was used to support **H**, rather than the *curl* of a scalar potential as done here.

However, the governing equations above are happily almost identical to those in [10] due to the natural symmetries present in the mathematical constructs—except for the transmission conditions (5f) and (5e) which have swapped roles.

While the $\nabla \cdot \mathbf{B} = 0$ gives rise to the Laplacian on the potential, and $\nabla \times \mathbf{H} = 0$ is identically satisfied when using a gradient formulation, the opposite is true when using the curl based forms above.

Similarly, the $\mathbf{H} \times \mathbf{n}$ and $\mathbf{B} \cdot \mathbf{n}$ boundary continuities exchange their roles working with either the scalar potential or its normal derivative over said boundary—which is also a reversal of the conditions found in the gradient formulation.

Because of these formulation symmetries, an almost identical time iterative scheme to that presented in [10] is adopted, with iteration equations to be summarized after the non-dimensionalization coming next, but with the further implementation details using piecewise linear and constant finite elements not repeated thereafter.

### 2.3. Non-Dimensionalization

With magnetic susceptibility, $\chi$, already unitless, the non-dimensionalization of (10) follows that done in previous work [10,12] with tildes on the corresponding (untilded) physical quantities to indicate dimensionless time $\tilde{t} \equiv t/\hat{t}$, density $\tilde{\rho} \equiv \rho/\hat{\rho}$, surface tension $\tilde{\gamma} \equiv \gamma/\hat{\gamma}$ and pressure $\tilde{p} \equiv p/\hat{p}$ where

$$\hat{t} = \sqrt{\frac{\rho\,a^3}{\gamma}}; \qquad \hat{\rho} = \frac{\nu\,\hat{t}}{a^2}; \qquad \hat{\gamma} = \frac{\nu\,a}{\hat{t}}; \qquad \hat{p} = \frac{\nu}{\hat{t}} \tag{7}$$

and the physical distance scale, $a$, may here be considered to be the radius of the sphere with equal volume to the toroid of interest.

With the 'Ohnesorge' number again providing a measure of non-dimensional viscosity by dividing the square root of the Weber number $We = (\rho\,u^2 a)/\gamma$ by the Reynold's number, $Re = (\rho\,u\,a)/\nu$

$$Oh := \frac{\sqrt{We}}{Re} = \frac{\sqrt{\rho\,u^2 a}}{\sqrt{\gamma}}\frac{\nu}{\rho\,u\,a} = \frac{\nu}{\sqrt{\gamma\,a\,\rho}} = \frac{\nu}{\sqrt{\tilde{\gamma}\,\hat{\gamma}\,a\,\tilde{\rho}\,\hat{\rho}}} = \frac{1}{\sqrt{\tilde{\gamma}\,\tilde{\rho}}}\frac{\nu}{\sqrt{\frac{\nu\,a}{\hat{t}}\,a\,\frac{\nu\,\hat{t}}{a^2}}} = \frac{1}{\tilde{\rho}} = \frac{1}{\tilde{\gamma}}$$

the non-dimensionalization is completed by just redefining the dimensionless magnetic Bond number, $Bo_m$, in terms of a new non-dimensional current, $i_0$, in the wire instead

$$Bo_m = \frac{a\,\mu_0\,H_0^2}{2\,\gamma} = \frac{a\,\mu_0\,I_0^2}{2\,\gamma\,a^2} = \frac{\mu_0\,I_0^2\,\kappa_0}{4\,\gamma} = i_0^2 \tag{8}$$

where $\kappa_0 = 2/a$ is just the mean curvature (the sum of the principle curvatures) of the sphere of radius $a$ with equivalent volume to the torus of interest.

Dropping all the tildes from here on then gives for the momentum Equation (5a):

$$\frac{1}{Oh}\left\{\frac{\partial \mathbf{u}}{\partial t} + (\mathbf{u} \cdot \nabla)\mathbf{u}\right\} = -\nabla\,p - \Delta\,\mathbf{u} \tag{9}$$

### 2.4. Time-Stepping Strategy

Using a plus superscript, $x^+$, to indicate a value that is to be computed at a particular timestep, and a minus, $x^-$, for its value at the previous timestep, and adopting an "Arbitrary Lagrangian–Eulerian" (ALE) approach [13], with a mesh velocity, $\mathbf{v}$, to keep the interior mesh nodes well positioned, the governing system may be decomposed into a few FEM solves and one BEM solve to be performed at each iteration:

$$\frac{1}{Oh}\frac{\mathbf{u}^+}{\Delta t} + \frac{1}{2}\Delta\mathbf{u}^+ = \quad \frac{1}{Oh}\frac{\mathbf{u}^-}{\Delta t} - \frac{1}{Oh}\big[(\mathbf{u}^- - \mathbf{v})\cdot\nabla\big]\mathbf{u}^- - \nabla p^- - \frac{1}{2}\Delta\mathbf{u}^- \qquad \mathbf{u}, p \in \Omega \quad (10\text{a})$$

$$\nabla\cdot\mathbf{u}^+ = \qquad\qquad\qquad\qquad\qquad\qquad 0 \qquad \mathbf{u}\in\Omega \quad (10\text{b})$$

$$\frac{\partial\mathbf{u}^+}{\partial n} = \qquad \frac{2\kappa}{Oh}\mathbf{n} + \frac{i_0^2}{Oh}\chi\Big[\chi\,(\mathbf{H}\cdot\mathbf{n})^2 + H^2\Big]\mathbf{n} \qquad on\ \Gamma \quad (10\text{c})$$

$$\alpha\,\phi^+ + \Delta\phi^+ = \qquad\qquad\qquad\qquad\qquad \alpha\,\phi^- \qquad \phi\in\Omega \quad (10\text{d})$$

$$\frac{\partial\phi^+}{\partial n} = \qquad\qquad\qquad\qquad \frac{\partial\psi^-}{\partial n} + \frac{\partial h_0}{\partial n} \qquad \phi,\psi\in\Gamma \quad (10\text{e})$$

$$\oint_\Gamma G\frac{\partial\psi^+}{\partial n}\,ds = \quad 2\,\pi\Big\{\frac{\phi^-}{(1+\chi)} - h_0\Big\} + \oint_\Gamma\Big\{\frac{\phi^-}{(1+\chi)} - h_0\Big\}\frac{\partial G}{\partial n}\,ds \qquad \phi,\psi\in\Gamma \quad (10\text{f})$$

Detailed discussion on the actual linear finite-/boundary-element discretisation, integration-by-parts, and implementation of the above iterative equations may be found in [10], and is not repeated here, but note the sign reversal on the magnetic forcing term in (10c) due to the magnetisable medium now being on the *outside* of the droplet/ambient-fluid interface $\Gamma$.

It was noted by a reviewer of the present work that the above decomposition into scattered, incident and total fields was very similar to that of early "reduced" scalar potential formulations, see [14] and references therein, where "the gradient of which is defined to be the field from the magnetized regions of the problem, that is to say, the total field diminished by the known source [incident] fields."

Interestingly, in regions containing magnetically permeable media a problematic "cancellation of the known source and calculated potential fields" was also identified that implied a formulation in terms of a *total* potential is necessary for such regions [14].

Now at first glance this looks problematic, because a scattered (or "reduced") potential for the permeable region here is unavoidable since the latter is unbounded and thus needs a boundary integral formulation whose operators can only support fields that decay towards infinity—something which a total field obviously does not do.

Fortunately, however, [14] then also further suggests that this cancellation problem in permeable regions applies only for magnetic scalar potentials, thus working with a magnetic *vector* potential (even when using but one scalar component of it) will hopefully neatly side-step any such issues in the now unbounded permeable region considered here because "a vector potential formulation, [...] does not suffer from [such] cancellation problems".

### 3. Results and Discussion

In Figure 1 (left) can be seen the initial volume torus mesh used for all the major calculations presented in the current work. It was generated using the *distmesh* algorithm [15] and has 4407 nodes and 24,699 tetrahedral elements, with major and minor radii of $R = 0.66$ and $r = 0.30$, respectively, giving an overall volume of about 0.1.

The exact radii chosen just come from a simple "parameter sweep" using the mesh generator over very many different possible values for both radii and desired mesh element diameter looking for a sufficiently dense mesh, but of manageable computational size, and with a good enough quality—as measured by the maximum volume ratio of any tetrahedron in the mesh to its respective circumsphere [16].

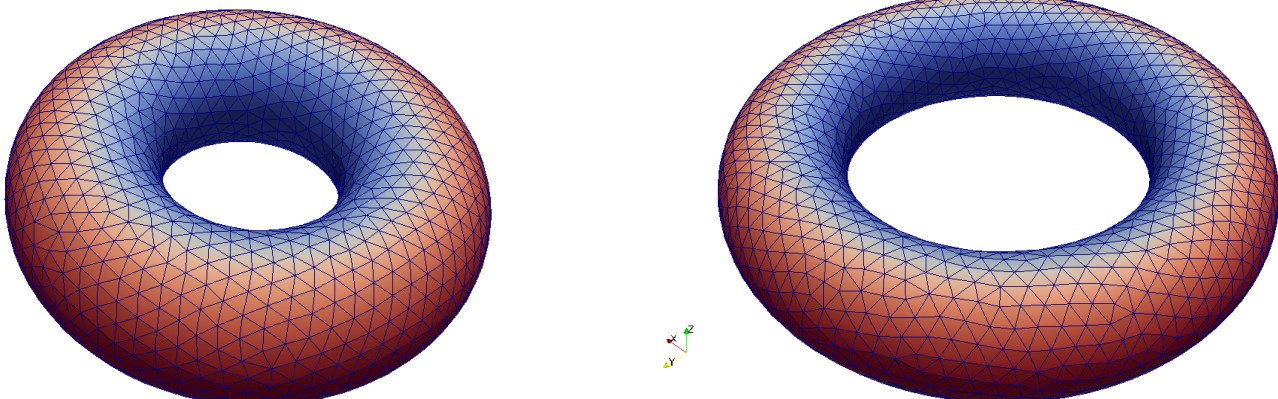

**Figure 1.** Initial "fat" $R = 0.66$ and $r = 0.30$ torus mesh used for most calculations (**left**) and alternative "thin" $R = 0.84$ and $r = 0.24$ mesh just for validation experiment (**right**—see text). Indicative colouring corresponds to MVP values.

Starting from this initial torus mesh, the iterative scheme described above, with a magnetic susceptibility of $\chi = 5$, was allowed to progress over non-dimensional time for different choices of (the square of) the non-dimensional current $i_0$ using a timestep $\Delta t = 0.01$ below which preliminary experiments indicated little improvement in overall solution stability.

The calculations would continue until either a given end time point of $t = 20$ was achieved, or a break-down of the calculations occurred if the electric current was either (a): insufficient to prevent the collapse of the whole toroidal drop naturally under the effects of surface tension, or (b): big enough to lead to an extreme expansion and certain loss of axi-symmetry—either of which would usually lead to a point of such poor mesh quality that the calculations would cease.

The changes to the *"proxi"*-major radius, $\tilde{R}$, of the torus over time for the different wire currents then tried may be seen in Figure 2, and are calculated from the torus volume and the maximum distance of its surface from its centre *on the assumption of a perfectly uniform minor radius*.

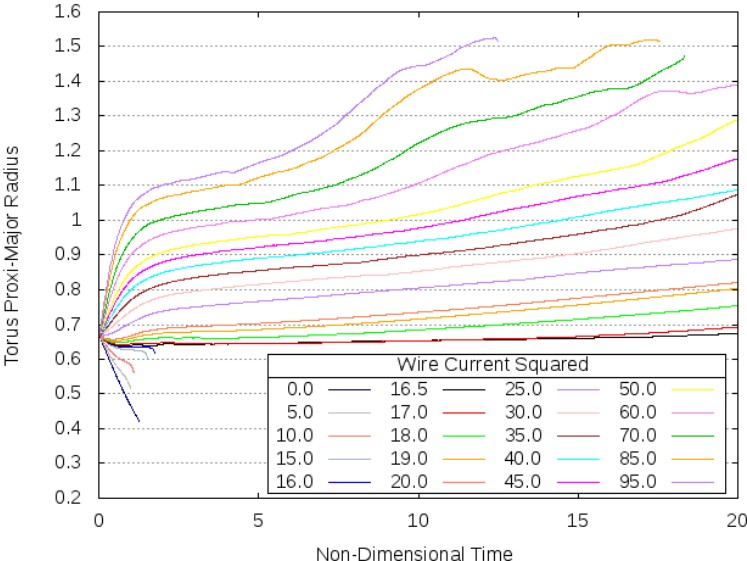

**Figure 2.** Plots of the proxi-major torus radii $\tilde{R}$ over non-dimensional time $t$ from initial state, $\tilde{R} \equiv R = 0.66$, for a selection of different wire currents, $i_0^2$.

Of course, this assumption of a perfectly circular minor cross-section is soon violated after most currents tested, as will be seen, however, this proxi-major radius does provide a readily calculable and convenient basic overall size measure for the torus which can easily identify rough stability and instability trends of its general form over time.

With convergence of the MVP solution happening within a few iterations of the start—as with the magnetic scalar potential (MSP) in [10]—it is safe to assume that the balancing acts between the magnetic and surface tension forces under-pinning the behaviours seen in Figure 2, commence almost immediately and drive all the motions from the very beginning.

Furthermore, as stated earlier, it is the possibility of a stable (or at least relatively stable) equilibrium shape that is of primary interest here, with the kinematics involved in actually achieving one being of less importance, and so a very large non-dimensional viscosity, or Ohnesorge number, of $Oh = 5$, was used in all the calculations for Figure 2 to effectively "stabilise" against any small irregularities in either mesh or local surface tension force calculations by heavily slowing down any adverse effects they might otherwise have over time.

Studying Figure 2, it is clear that the most "stable" current of those tried is found around $i_0^2 = 16.5$, for which the proxi-major radius remains broadly the same for most of the time—and also quite close to the initial value. Smaller currents cannot prevent the natural torus collapse under the effects of surface tension, while larger currents just cause ever greater expansions over time.

So, to give some sense of a physical real-world problem, for a non-magnetic toroidal fluid droplet surrounded by a light hydrocarbon oil-based ferrofluid with the magnetic susceptibility $\chi = 5$ used for the present results and with, say, the same volume as a 3 millimetre radius sphere and a surface tension coefficient with the ferrofluid medium of 30 dynes per centimetre, Equation (8) would suggest

$$
\begin{aligned}
I_0^2 &= \frac{2\,i_0^2\gamma\,a}{\mu_0} \\
&= \frac{2 \times 16.5 \times 30 \times 10^{-5}\,N\,100\,m^{-1}\,3 \times 10^{-3}\,m}{4\,\pi \times 10^{-7}\,J\,A^{-2}\,m^{-1}} \\
&= \frac{2 \times 16.5 \times 30 \times 3}{4\,\pi} \times 10\,A^2 \\
I_0 &\approx 48.6\,A
\end{aligned}
$$

Thus, a current of about 50 Amps would be needed in a perpendicular wire through the centre of the torus-shaped droplet to keep it quasi-stable in the SI system used here, where the magnetic permeability is expressed in Joules, $J$, per Ampere, $A$, squared per meter, $m$,—equivalent to "Henrys" per meter—and one dyne is just $10^{-5}$ Newtons, $N$.

Given the proximity of the stable proxi-major radius to its starting value, it was also thought prudent to just check that the major/minor radius ratio $R/r$ of the starting torus was not having an undue influence on the results—which, of course, should only depend on the overall *size* of the drop.

Thus, just for this purpose, a small selection of the wire electric current values were also tried with a thinner initial torus mesh with a major/minor radius ratio of $R/r = 0.84/0.24$ (and slightly fewer elements at 24,003), see Figure 1 (right), which was then linearly scaled geometrically to have exactly the same volume as the first.

The results from using both this "thin" torus, and the original "fat" one, can be seen in Figure 3—with a logarithmic time scale introduced just to clarify the two different starting points given the rapid early changes—and do indeed suggest that the initial radii ratio has but a limited influence.

Returning to the results of Figure 2, what is very interesting about the most stable toroidal droplet configuration that has been found at $i_0^2 = 16.5$ is the complete loss of circular minor cross-section from the initial form, as can be seen developing over time in the

sequence of cross-sections shown in Figure 4—to be viewed chronologically in descending vertical column order, as per normal columnised text.

The unusual "egg" shaped stable minor cross-section that can be seen developing in Figure 4 is quite striking, but starts to make sense when one considers the balance of surface forces that is needed to achieve such a relatively stable state.

The inside edge of the torus is inevitably closer to the current carrying wire located at the torus centre than the outer edge, thus the (annular) magnetic field there will be correspondingly stronger too, and the surface magnetic forces arising from this field equally so.

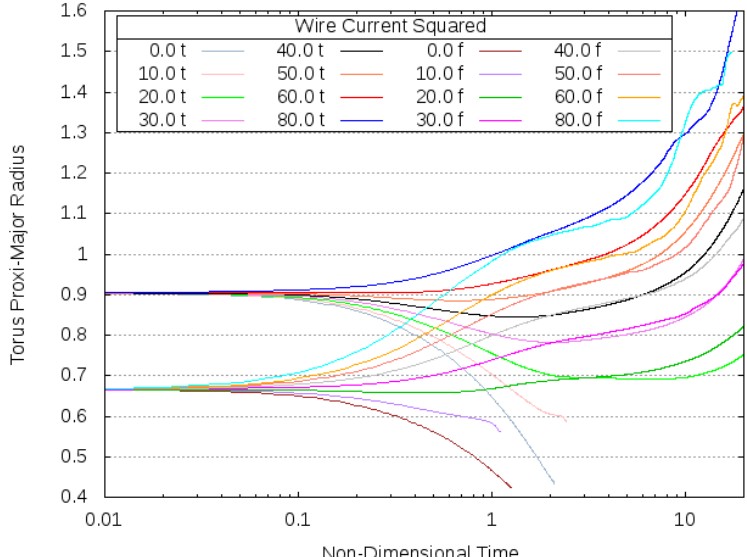

**Figure 3.** Plots of the proxi-major torus radii $\tilde{R}$ over non-dimensional time $t$ from two different initial states "f" and "t", with respectively $\tilde{R} \equiv R = 0.66$ and $\approx 0.9$ (scaled out from 0.84 to give same torus volume—see text) for a selection of different wire currents, $i_0^2$.

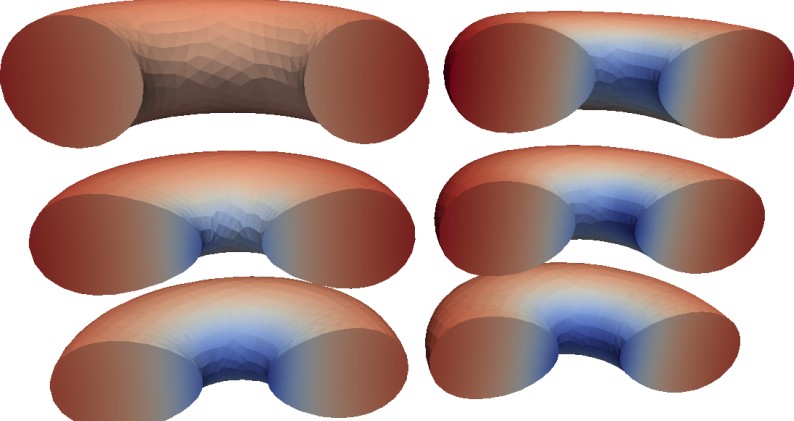

**Figure 4.** Cross-section sequence over time from the initial torus mesh used for the calculations (**top left**) to the quasi-steady state form found using a wire current of $i_0^2 = 16.5$ to the same scale (**bottom right**). Indicative colouring corresponds to MVP values. Note the distinctive "egg" shaped minor cross-section that develops over time.

However, for stability of the *whole* torus shape these magnetic surface forces need to be perfectly balanced out by the surface tension forces *everywhere*.

By adopting an egg cross-sectional shape—with the "pointy" end directed towards the torus centre—this can be achieved, however, because the smaller radii of curvature

on the inside edges will naturally strengthen the surface tension forces there, which are inversely directly proportional via the constant surface tension coefficient.

Note that this egg-like cross-section can also develop to differing degrees with current values below that of the roughly "stable" form—except of course for the zero current case as seen in Figure 5—and thus effectively further limits just how low the proxi-major radii can go in Figure 2 before the corresponding meshes degenerate as the torus "hole-in-the-middle" fills-in and disappears.

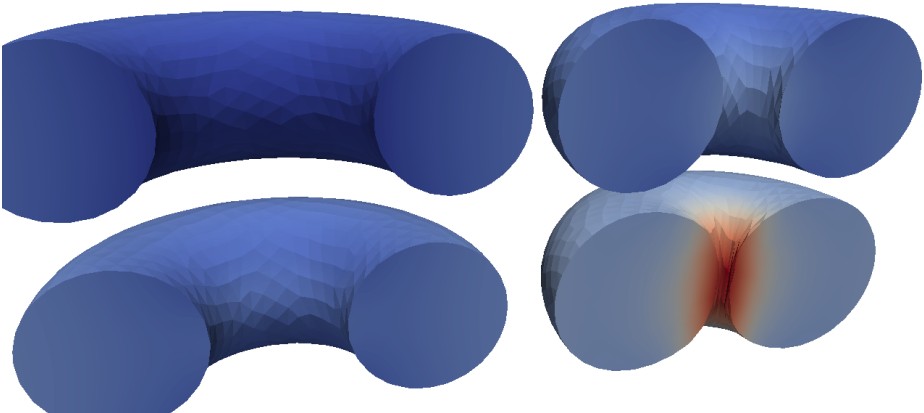

**Figure 5.** Cross-sections of the initial torus mesh used for the calculations (**top left**) evolving to that of the almost collapsed drop (**bottom right**) with no wire current to the same scale. Indicative colouring corresponds to MVP values. Note the broadly continued circular minor cross-section throughout the collapse owing to the absence of any magnetic field.

Naturally, for a given overall torus volume and maximum distance of the surface from the centre, the greatest hole diameter (and thus usually the least degenerate mesh) is achieved when the minor cross-section is circular, see Figure 5 (bottom right), and thus it is the zero current case $i_0^2 = 0$ in Figure 2, which shows by far the lowest proxi-major radii achieved in all the experiments of about $\tilde{R} = 0.42$.

Now for wire currents $i_0$ greater than that of rough stability, it is suggested here that any egg shaped minor cross-section is progressively less effective at counter-acting the differential magnetic field strengths at the inside and outside edges of the torus as the electric current strengths increase, thus allowing the former to expand ever outwards, see Figure 6 for the most extreme $i_0^2 = 95$ case tried.

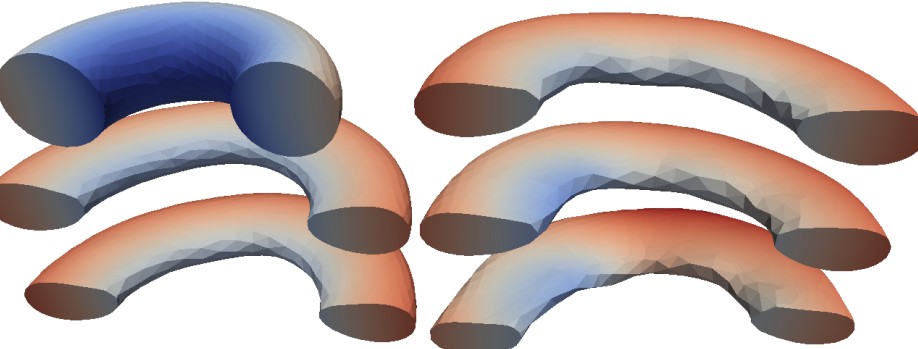

**Figure 6.** Cross-sections of the initial torus mesh used for the calculations (**top left**) expanding-out over time to one of the most extreme shapes (**bottom right**) with a wire current of $i_0^2 = 95$. Indicative colouring corresponds to MVP values. Note the loss of axi-symmetry towards the end.

As seen in this figure, while the egg outline does develop at first, the torus, however, continues to expand, with the ever smaller minor cross-section becoming an ever more distorted egg shape as it does so, and evidently failing to balance the ever more unequal magnetic forces felt on the inside and outside edges of the torus as it gets thinner and thinner.

It must be said, however, that a certain loss of axi-symmetry is present in the latter stages of many of these more extreme results with the larger wire currents, especially when the proxi-major radii get large, which could be influencing the above interpretation.

Now these asymmetries may or may not have their *origins* in the assumptions made for the MVP formulations described above, with some of the 'z'-derivatives of the MVP perhaps becoming less negligible with new distortions over time, but they could start to test those assumptions either way, and this should be bourne in mind when considering the results.

However, adopting these assumptions has now at least shown the possibility for broadly stable, non-magnetic, toroidal droplets about an azimuthal magnetic field to be maintained within a surrounding ferrofluid medium, and this presents some interesting possible applications.

If the droplet and surrounding matrix fluids are immiscible with different densities, as they easily could be, then the exact line of motion of such toroidal droplets up or down (depending on their relative buoyancy in a gravitational field, say), could be precisely controlled by effectively "threading" them onto an electric current carrying wire like beads on a string.

With surface tension always trying to collapse the torus to a sphere, and the magnetic forces from the wire current always opposing this collapse, but weakening with distance from the wire, the natural tendency would perhaps be for the toroidal drops to stay centred *and* perpendicular with respect to this wire as they move along it.

To quickly test this out, two final very simple numerical experiments were performed using the most stable $i_0^2 = 16.5$ wire current found above.

The first involved tilting the initial torus mesh by 45 degrees to the direction of the wire (always coincident with the 'z'-axis), while the second displaced the whole initial mesh by a distance of about 80% of the radius of the inner torus "hole" perpendicularly away from the wire—thus avoiding the wire itself (and hence singular magnetic fields) ever being actually within the computational domain.

The results of these two experiments may be seen in Figure 7 (left) and (right) respectively, and show a very rapid—relative to the egg cross-section formation—effective "correction" of both initial tilts and initial displacements back to the wire centred and perpendicular over time, thus confirming suspicions.

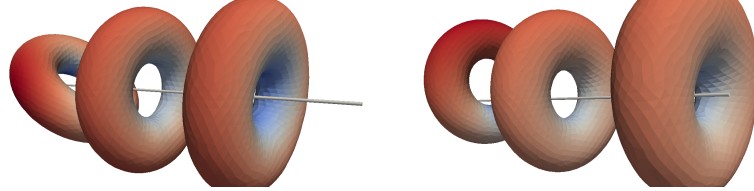

**Figure 7.** Snap-shots of the computational torus with a wire current of $i_0^2 = 16.5$ showing "correction" of both an initial tilt (**left**) and an initial perpendicular displacement (**right**) over time, with a very thin cylinder marking the z-axis for reference. Note the almost complete corrections in both tilt and displacement before the development of the egg minor cross-section does much change to the centre hole diameters.

However, some physical experiments might also be desirable to fully confirm that the toroidal drops *would* naturally correct any angled tilt away from the wire perpendicular and any displacement of its centre from the wire itself as suggested here, but otherwise such a precise control of toroidal drop positioning could be extremely useful.

Furthermore, of course simply switching off the stabilising magnetic field by turning the electric current in the wire off at any moment could also have applications by automatically triggering an instant collapse in all the toroidal droplets (or even bubbles···) travelling or aligned along it at the time, allowing for the controlled release of sound waves, for example, or the initiation of mixing processes as required.

The size of such toroidal droplets could even be used as a proxi for identifying the current actually flowing within a wire, giving an easy to use optical measure, although this might also require a knowledge of the time involved too given that "stability" is not found with all wire currents.

Finally, it should be noted that while the assumption of negligible 'z'-variance of the MVP, made in the formulations above, $\phi_z \approx \psi_z \approx 0$, could be considered quite a severe constraint on the modelling, *at no point are zero 'z'-derivatives actually enforced directly in a Dirichlet manner*—which would of course then force the MVP solution to be truly two-dimensional.

Thus, it is hoped that the iterative scheme used for the results presented here allows for a relatively negligible but still sufficient 'z'-variability of the magnetic field, via the MVP, to give at least an approximation to the true, three-dimensional, real-world behaviour—and at least enough to inspire some future physical and further numerical experiments in this direction.

## 4. Conclusions

The relatively stable toroidal shape of a non-magnetic droplet about an azimuthal magnetic field within an immiscible surrounding ferrofluid has been established via a coupled finite element/boundary element numerical simulation.

The loss of a perfectly circular minor cross-section has been noted in the stable shape, with instead an "egg" shaped profile allowing surface tension forces to locally balance magnetic surface forces that increase towards the torus centre.

Applications have been suggested for the precise controlled motion of toroidal droplets or bubbles, due to buoyancy in a gravitational field, along an electric current carrying wire generating the required azimuthal magnetic field. Stability corrections against both tilting and perpendicular displacement from the wire have also been suggested by numerical calculations.

Finally, it should be noted that future numerical investigations could benefit from the use of a true axi-symmetric formulation, to both avoid non-axisymmetric perturbations polluting the solution and for speed of computation allowing the easy use of larger mesh densities—though convergence of an MVP over a much more restricted boundary might be more difficult.

**Funding:** This research received no external funding

**Data Availability Statement:** All the computer simulations presented in this work were based on an open source *DUNE* software module written in the C++ programming language [17].

**Conflicts of Interest:** The author declares no conflict of interest.

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
