# Peer review of "Quasi-Stable, Non-Magnetic, Toroidal Fluid Droplets in a Ferrofluid with Annular Magnetic Field"

_2673-8724, doi:10.3390/magnetism2040027_

Round 1

Reviewer 1 Report

To my mind this work can present interest to some part of ferrofluid community, however not to a wide part. In the other points I consider this work as not badly maid theoretical study of the considered problem. I have only some methodical remarks.

1. Please, formulate clearly in Introduction the considered physical problem and the novelty of your consideration.

2. Traditionally, the symbol "nu" marks kinematic, not dynamic viscosity. The usage of the non traditional notation can be inconvenient for readers.

3. Figures 2,3 are given in the dimensionless units. Please, discuss the corresponding dimensional radius and time, so that the reader could understand the physical situation, not only the mathematical results.

Author Response

Author's Reply (Reviewer 1)

The author gratefully acknowledges the comments and suggestions by the reviewer and would like to respond to the specific remarks in the order they were made as follows:

1. Please, formulate clearly in Introduction the considered physical problem and the novelty of your consideration.

>>>>> The problem is clearly stated in the abstract, with the novelty of a toroidal droplet with the fluid/ferrofluid regions reversed to that previously considered stated in the introduction.

2. Traditionally, the symbol "nu" marks kinematic, not dynamic viscosity. The usage of the non traditional notation can be inconvenient for readers.

>>>>> Priority was given to the use of “mu” for magnetic permeability, rather than use it for the traditional dynamic viscosity, so “nu” was used for the dynamic viscosity instead because it does at least have a viscosity association.

>>>>> A small in-line comment on this has been included on its first introduction on page 1, line 38 to alert the reader to this.

3. Figures 2,3 are given in the dimensionless units. Please, discuss the corresponding dimensional radius and time, so that the reader could understand the physical situation, not only the mathematical results.

>>>>> A dimensioned example problem for a real world physical situation has now been suggested on page 8, line 213.

Reviewer 2 Report

The paper reveals the existence of quasi-stable toroidal drops. A non-magnetic fluid, located in the volume of a magnetic fluid, can take such a form under the influence of the magnetic field of a current-carrying conductor and surface tension. Unlike other studies of toroidal drops ([7] – [9] from list of References), the author considers a free drop not in contact with a solid surface or the surface of another drop lying on a solid surface.

The system of equations, the solution method and the presented results are well-founded. However, the author gives a solution obtained on a grid containing 4407 nodes and 24699 tetrahedral elements, but does not compare with solutions obtained on other grids. Therefore, it remains unclear whether the grid used is sufficient for the reliability of the results obtained.

From the data presented by the author (Figures 2 and 3) it does not follow that at some value of the current in the conductor, the toroidal drop is absolutely stable. Even on the curve for i02 = 16.5 (Figure 2), the radius does not remain constant. The author uses the term “relatively stable”, but the unambiguous statement “stable” is included in the title of the article, although the statement about the absolute stability of the toroidal form has not been proved by the author. Therefore, I would recommend using the term “quasi-stable” in the title of the article.

The results obtained are certainly interesting and may be published in the journal Magnetism.

Author Response

Author's Reply (Reviewer 2)

The author gratefully acknowledges the comments and suggestions by the reviewer and would like to respond to the specific remarks in the order they were made as follows:

The system of equations, the solution method and the presented results are well-founded. However, the author gives a solution obtained on a grid containing 4407 nodes and 24699 tetrahedral elements, but does not compare with solutions obtained on other grids. Therefore, it remains unclear whether the grid used is sufficient for the reliability of the results obtained.

>>>>> Further studies in this direction might well be pursued in future with more resources available, especially if the present work stimulates some physical experimental results in the area to compare with.

From the data presented by the author (Figures 2 and 3) it does not follow that at some value of the current in the conductor, the toroidal drop is absolutely stable. Even on the curve for i02 = 16.5 (Figure 2), the radius does not remain constant. The author uses the term “relatively stable”, but the unambiguous statement “stable” is included in the title of the article, although the statement about the absolute stability of the toroidal form has not been proved by the author. Therefore, I would recommend using the term “quasi-stable” in the title of the article.

>>>> Good suggestion: “Quasi-Stable” now appears in the title and elsewhere.

Author Response

Author's Reply (Reviewer 3)

The author gratefully acknowledges the comments and suggestions by the reviewer and would like to respond to the specific remarks in the order they were made as follows:

R1:  Page2, line 52.

Here is stated that the magnetic stress tensor is present only or mainly in a ferrofluid region. This can be assumed when the ferrofluid relative permeability is much greater than the unit, but it usually ranges from 1.3 to 5. Please state the used value of the ferrofluid relative permeability in the revised version of the article.

>>>>> The relative permeability is trivially defined by the magnetic susceptibility (but an extra in-line equation has now been included on page 3, line 102) which is stated as being equal to 5 at the beginning of the results section.

   R2: Page 2, Line 72.

The magnetic formulation is based on the total and scattered magnetic potential. Is there any notable difference between the formulation through the total and reduced magnetic potential introduced in refA?

refA: J. Simkin, C. W. Trowbrodge, “On the use of the total scalar potential in the numerical solution of field problems in electromagnetics”, International Journal for numerical Methods in Engineering, Vol. 14, No. 3, pp. 423-440, 1979

The reviewer would appreciate it if the author could give some words on the subject.

>>>>> Well spotted !!!    Yes, there is a great deal of similarity, however, I have to use a scattered (or “reduced”) potential for the exterior region because it is unbounded and thus needs an integral formulation whose operators can only support fields that decay towards infinity, something which the total field obviously does not do.

>>>>> But if you are concerned about accuracy in the exterior permeable region, because your ref. suggests the problem of cancellation of known source and calculated potential fields in permeable regions applies only for magnetic scalar potentials, because I’m working with a magnetic VECTOR potential (even though I’m only using one scalar component of it) any such cancellation in the exterior region should not be problematic since, as your ref. also says: “a vector potential formulation, […] does not suffer from [such] cancellation problems”

>>>>> Some paragraphs describing this have now been added at the end of the formulations.

R3: Page 4, Eq. 5d-5h

Please explain to the reader what Γ (gama) stands for.

>>>>> Gamma is already stated as being the surface of the droplet on page 3, line 105, but an extra mention of it has now been included on page 2, line 40.

R4: Figure 1

Coordinate labels should be more significant. A similar issue is with figures 4 – 7.

>>>>> These figures are intended for qualitative analysis only, quantitative analysis (and hence the need for coordinate labels) is kept to the graphical data presented in the remaining figures.